# Acceptability and feasibility of home and hospital follow-up in Burkina Faso and Guinea: A mixed-method study among patients of the COVID-19 Coverage-Africa clinical trial

**Mélanie Plazy**[1]☯*, **Marie-Hélène Doucet**[1]☯, **Christine Timbo Songbono**[2,3], **Anselme Sanon**[4], **Bamba Issiaka**[4], **Caroline Martin**[3], **Inès Da**[4], **Anthony L'hostellier**[1], **Olivier Marcy**[1], **Denis Malvy**[1,5], **Armel Poda**[6], **Alexandre Delamou**[2], **Abdramane Berthé**[4], **Joanna Orne-Gliemann**[1]

1 University of Bordeaux, National Institute for Health and Medical Research (INSERM) UMR 1219, Research Institute for Sustainable Development (IRD) EMR 271, Bordeaux Population Health Research Centre, Bordeaux, France, 2 African Centre of Excellence in the Prevention and Control of Communicable Diseases (CEA-PCMT), Faculty of Sciences and Health Techniques, Gamal Abdel Nasser University, Conakry, Republic of Guinea, 3 The Alliance for International Medical Action (ALIMA), Conakry, Republic of Guinea, 4 Muraz Centre, Department of Public Health, Bobo-Dioulasso, Burkina Faso, 5 Division of Tropical Medicine and Clinical International Health, Department of Infectious Diseases and Tropical Medicine, CHU Pellegrin, Bordeaux, France, 6 Superior Institute of Health Sciences, Nazi Boni University, CHU Sourô Sanou, Bobo Dioulasso, Burkina Faso

☯ These authors contributed equally to this work.
* melanie.plazy@u-bordeaux.fr

**Data Availability Statement:** Study data cannot be made publicly available. The ANRS COV33

## Abstract

Patient experiences and perspectives on trial participation and follow-up may influence their compliance with research procedures or negatively impact their well-being. We aimed to explore the acceptability and feasibility of home-based and hospital-based follow-up modalities among COVID-19 patients enrolled in the ANTICOV ANRS COV33 Coverage-Africa trial in Burkina Faso and Guinea. The trial (2021–2022) evaluated the efficacy of treatments to prevent clinical worsening among COVID-19 patients with mild to moderate symptoms. Patients were either based at home or hospitalized, as per national recommendations, and followed-up through face-to-face visits and phone calls. We conducted a mixed-methods sub-study administering a questionnaire to all consenting participants and individually interviewing purposively selected participants. We performed descriptive analyses of Likert scale questions for the questionnaires and thematic analysis for the interviews. We conducted framework analysis and interpretation. Of the 400 trial patients, 220 completed the questionnaire (n = 182 in Burkina Faso, n = 38 in Guinea) and 24 were interviewed (n = 16 and n = 8, respectively). Participants were mostly followed-up at home in Burkina Faso; all patients from Guinea were first hospitalized, then followed-up at home. Over 90% of participants were satisfied with follow-up. Home follow-up was considered acceptable if (i) participants perceived they were not severely ill, (ii) it was combined with telemedicine, and (iii) the risk of stigma could be avoided. Hospital-based follow-up was viewed as a way to prevent

COVERAGE Africa study is sponsored by the ANRS MIE (French Agency for Research on AIDS and viral hepatitis / Emerging Epidemic Diseases). We are therefore obliged to abide by French and European legal restrictions such as French law no. 78-17 of January 6, 1978 relating to data processing, files and freedoms usually called "Loi informatique et libertés" and European law "GDPR" General Data Protection Regulation that govern data sharing and stipulate that data cannot be shared in open access to an undeclared third party if study participants have not been informed and did not give their consent for it. Thus, persons wishing to have access to the interviews or questionnaire data may submit a request to the study sponsor (ANRS, clinique@anrs.fr), who will take steps to assess whether the request can be fulfilled. That said, all quantitative and qualitative (verbatim extracts) data allowing to answer our study questions are presented in the paper.

**Funding:** This work was supported by ANRS | MIE (sponsor, ANRS COV33) and by the global health agency Unitaid as part of ACT-A. The ANTICOV Study is also funded by the German Federal Ministry of Education and Research (BMBF) through KfW, and received additional support from the European & Developing Countries Clinical Trials Partnership (EDCTP) – under its second programme supported by the European Union with additional funding from the Swedish government – the Starr International Foundation and the Stavros Niarchos Foundation (SNF). The funders had no role in study design, data collection and analysis, decision to publish, or preparation of the manuscript.

**Competing interests:** The authors have declared that no competing interests exist.

contamination of family members, but could be badly experienced when mandatory and conflicting with family responsibilities and commitments. Phone calls were seen as reassuring and as a way to ensure continuity of care. These overall positive findings support the development of home-based follow-up for mildly ill patients in West-Africa, provided that both emotional and cognitive factors at individual, familial/inter-relational, healthcare and national levels be addressed when planning the implementation of a trial, or developing any public health strategy.

## Introduction

The COVID-19 pandemic has challenged health systems and societies over the world. Despite the relatively low number of confirmed cases [1], the virus causing COVID-19 has spread widely on the African continent, with an estimated rate of two-thirds (65.1%) of Africans having been exposed to the SARS-CoV-2 virus [2].

The response of national health authorities to the COVID-19 pandemic and crisis, whether in terms of prevention, management and treatment guidelines, has differed between African countries and has evolved over time. The previous Ebola Virus epidemic in Guinea, Liberia, and Sierra Leone played an important role in triggering partial or full lockdowns of uninfected persons and/or isolation of people who tested positive for SARS-CoV-2 during the current COVID-19 pandemic [3, 4]. In some countries for instance, all people with COVID-19 were immediately and strictly isolated in healthcare facilities, while in others, only people with severe COVID-19 symptoms were hospitalized and milder cases were asked to self-isolate at home [5]. In addition, mainly because this emerging epidemic disease was not well understood at first and effective treatments were lacking back in 2020, many people expressed fear of contamination and death [6–8]. Stigmatization towards those sick and recovered from COVID-19 has thus been reported [9–12], especially in settings that have faced the recent Ebola epidemic [13].

A large number of clinical trials have been conducted over the world [14] to find safe and efficient treatments for COVID-19, but very few have been conducted in Africa [15]. While some hospital-based trials aimed to reduce the risk of death among hospitalized COVID-19 patients with severe symptoms, others aimed to prevent clinical worsening among ambulatory COVID-19 patients with mild and moderate symptoms [16]. Such patients do not require specialized inpatient care, and can therefore be followed in outpatient units or at home.

Participating in a clinical trial often requires extensive follow-up beyond the disease itself, which may be experienced differently depending on whether participants are followed-up at home or in a healthcare facility. Their experience may consequently negatively affect their well-being and ultimately impact the implementation and outcomes of the trial. Conducting a therapeutic trial on COVID-19 among mildly symptomatic patients may add further particularities to their trial experience, including if they are followed-up at home. They may not feel sick, which may reduce their compliance to treatment and research procedures; they may feel constrained in their daily routine by the strict requirements of research procedures; or they may fear the indirect disclosure of their COVID-19 disease to their entourage/neighbors when being visited by the medical team. There is scarce literature exploring the experiences and points of view of patients followed-up once they are enrolled in a therapeutic clinical trial for an emerging infectious disease. Only one qualitative study was found, which described experiences and challenges in participating in a hospital-based COVID-19 trial among patients with severe symptoms in Austria [17]. Outside of the infectious disease field, some studies assessed the implementation of home-care strategies for diverse indications including chronic

obstructive pulmonary disease [18], chemotherapy for cancer [19], and care following knee or hip replacement [20]. However, no literature exploring the experiences and points of view of patients with mild symptoms participating in a clinical trial for an emerging infectious disease was found, whether followed-up at home or in a hospital, including in the African context.

We sought to explore the acceptability and feasibility of home-based and hospital-based follow-up among patients with mild to moderate symptoms enrolled in a COVID-19 clinical trial in Burkina Faso and Guinea.

## Materials and methods

### Study setting: The ANTICOV ANRS COV33 Coverage-Africa

This research was conducted as part of the ANTICOV ANRS COV33 Coverage-Africa (ClinicalTrials.gov Identifier: NCT04920838), a clinical trial, sponsored by ANRS-MIE and nested in the 01-COV ANTICOV platform trial, that aimed at evaluating the efficacy of repurposed drugs to prevent clinical worsening among COVID-19 patients with mild to moderate symptoms. The trial was implemented in the West-African urban contexts of Conakry (Guinea, 12 Apr 2021–30 Dec 2021), Ouagadougou and Bobo-Dioulasso (Burkina Faso, 29 July 2021–21 Nov 2022). This randomized clinical trial tested several treatments, including inhaled and oral drugs (in tablet form). Patients eligible for trial enrolment were at an early stage of their disease (less than 7 days of symptoms), had symptomatic but non-severe disease, i.e. with peripheral oxygen saturation equal of above 94% as measured by pulse oximetry, and were 40 years of age or older, or 18 years of age or older with comorbidity including obesity, treated hypertension or treated diabetes mellitus.

Recommendations for COVID-19 management have differed between countries. In Burkina Faso, individuals who tested positive for SARS-CoV-2 and had mild to moderate symptoms have been asked to stay at home: they can either be visited at home by mobile teams, or be followed-up in outpatient settings (i.e. going back and forth to a nearby health facility for their follow-up). Hospitalization has been restricted to patients with more severe symptoms or at risk of clinical worsening. In Guinea, all patients (both symptomatic and asymptomatic) have been hospitalized in dedicated treatment centers; patients were discharged home when they tested negative for SARS-CoV-2.

Trial procedures regarding patient enrolment and follow-up were defined in line with these national COVID-19 management recommendations prevailing during the study period. Trial participants were followed-up for 28 days. Clinical follow-up included: 1) face-to-face assessments (at inclusion and on days 7, 14 and 21), either in a hospital, at home or in a local outpatient clinic; and 2) daily follow-up phone calls for three weeks except at days 7, 14 and 21, with a final end-of-study phone call on day 28; patients could contact the trial medical team whenever they needed to.

### Study design and participants

As part of the trial, we conducted a mixed-methods sub-study following a convergent parallel design, aiming at assessing the acceptability and feasibility of trial implementation (thereafter called the 'Accept study'), from the point of view of different stakeholders. The present paper focuses on patient perspectives regarding follow-up in the trial, combining quantitative and qualitative data collected from the first enrolment in April 2021 to the end of February 2022. The quantitative component aimed at measuring, in an exhaustive and standardised way, the level of patients' satisfaction and agreement regarding specific statements, and facilitating comparisons between the different follow-up models and between countries. The qualitative component aimed at exploring in depth participants' individual experiences, views, and

feelings about these modalities of follow-up. Combining both quantitative and qualitative data aimed at providing a comprehensive assessment of the acceptability and feasibility of the different modalities of follow-up from patients' perspectives. The 'Accept study' was introduced to all trial patients, using an information sheet presented by the investigator during trial enrolment; those who agreed to participate in the sub-study signed a consent form. All consenting participants were invited to respond to a structured questionnaire. A sub-sample of participants was also invited to consent to individual in-depth interviews. We purposively selected these patients to maximize heterogeneity in their profiles and experiences within the trial [21], mainly according to the following criteria: gender; age; schooling level; follow-up setting (home or hospital); and treatment administration method (inhalation or oral tablets).

## Conceptual framework

We assessed participants' experiences and views about the acceptability of the different follow-up models, referring to their satisfaction and subjective assessment of the appropriateness of these models; this evaluation was based on both cognitive and emotional responses. To do so, we used Sekhon's Theoretical Framework of Acceptability [22], which includes seven domains such as the affective attitude (feeling) towards the intervention–in this case, the follow-up model–, the perceived burden for participating in the intervention, and the perceived effectiveness of the intervention. In addition, we investigated one specific aspect of the concept of feasibility [23], that is participants' views about the contextual conditions that made a follow-up model possible for them or not. During data analysis and interpretation, we used the Socio-ecological Model [24], to make sense of our results according to the following levels: individual (i.e. participant's experience or views concerning him/herself only); familial and inter-relational (i.e. relative to the participant's family members or entourage/neighbors); healthcare (e.g. relating to aspects concerning healthcare providers' work); national (i.e. regarding national policies or recommendations). Finally, participants' narratives were analyzed in the light of the sociocultural context in which they live, including the fact that West African societies are generally characterized by a culture of interdependence among family/community members [25, 26], as opposed to individualism.

## Data collection

Data for the Accept study were collected by phone (mainly to avoid the risk of COVID-19 transmission) by interviewers who were independent from the trial medical team, and who received extensive training on the objectives of the study and data-collection tools.

For the quantitative component, a 30-minute questionnaire was administered to patients 8 days (D8) after their inclusion in the trial. The questions addressed the following: socio-demographic information, perceptions regarding COVID-19, circumstances of the COVID-19 diagnosis, lived experience and satisfaction with follow-up within the trial, contacts with the medical team, treatment experience, perceptions of clinical trials, mental health, and contact with family and entourage. Satisfaction of a specific follow-up model was assessed only among those who reported having experienced that said model. Satisfaction questions were phrased as statements, with which participants could agree or not, using a five points ordinal Likert scale. Interviewers entered responses directly on a tablet dedicated to the study, using the Research Electronic Data Capture (REDCap) software.

For the qualitative component, two semi-structured individual interviews were conducted for each selected participant: 4 days (D4), and between 14 and 21 days (D14-21) after the trial enrolment. This two-step process aimed at preventing possible fatigue associated with a long interview over the phone (vs in person). D4 interviews lasted an average of 35 minutes, and

D14-21, 38 minutes. The interview guide explored at D4: experience of COVID-19 disease; beliefs regarding COVID-19; experience of taking part in the clinical trial; at D14-21: experience and points of view of the different types of follow-up; and consequences of the trial participation on their life. Data were audio recorded.

Study tools were designed in French and, when necessary, questions were translated into local languages and/or adapted to local culture to maximize participants' understanding [27]. In particular, key words and expressions were pre-translated into the main local languages to ensure maximum standardization in data collection. Interviews were conducted in French (n = 21) or in a local language (Guinean local languages: Soussou n = 1, Pular n = 1; Burkina Faso local language: Dioula n = 1), following participant's preference, and were thereafter transcribed into French. The quotes presented in this paper were translated into English.

### Data analysis

We analyzed participants' experiences and perceptions of being followed-up face-to-face and on the phone, while they were based at home or hospitalized. We did not investigate their experiences and views of being followed-up in outpatient settings as this modality was not part of the original trial implementation plan.

For the quantitative component, we performed descriptive analyses of Likert scale questions stratified by country and excluding missing values, using R software version 4.1.0.

For the qualitative component, we performed a classical thematic analysis [28], consisting of the following steps: 1) a codebook was developed, based on the themes of the interview guides as well as on the key elements relating to acceptability and feasibility of home, facility-based, and telephone follow-up modalities; 2) all transcripts were read, and then deductively coded using the MAXQDA qualitative data analysis management software (version 2022); 3) recurring sub-themes were identified and summarized.

Both quantitative and qualitative findings were then integrated for joint analysis and triangulation of all results. Our mixed-methods analysis aimed at revealing underlying meanings of participants' views and experiences, in context. To do so, we performed framework analysis, a systematic thematic analysis method which consists of managing and organizing results into a matrix, facilitating the identification of patterns (tendencies) in the data [21, 29, 30]. We conducted 3 phases of results classification: first within a three-dimensional matrix, comprising the acceptability and feasibility dimensions [22, 23], for each follow-up modality (home, hospital, telephone), and each country (S1 Table); then distinguishing between cognitive and emotional responses [22], i.e. primarily based on reason or emotions; finally according to the levels of the Socio-Ecological Model [24]. Overall, we interpreted the results taking into account the sociocultural contexts of Burkina Faso and Guinea. Results and interpretations were validated by the co-researchers in Burkina Faso and Guinea, acting as "cultural brokers" (i.e providing validation of cultural meanings) [31].

### Ethics

The Coverage-Africa study was approved by the National Ethics Committee for Health Research in Burkina Faso (No2021-02-047) and the National Ethics Committee for Health Research of the Republic of Guinea (No043/CNERS/21).

## Results

### Description of study participants

As of February 28, 2022, of 548 persons eligible for inclusion in Burkina Faso and Guinea, 400 were enrolled in the trial, of whom 334 consented to participate in the Accept study (Fig 1).

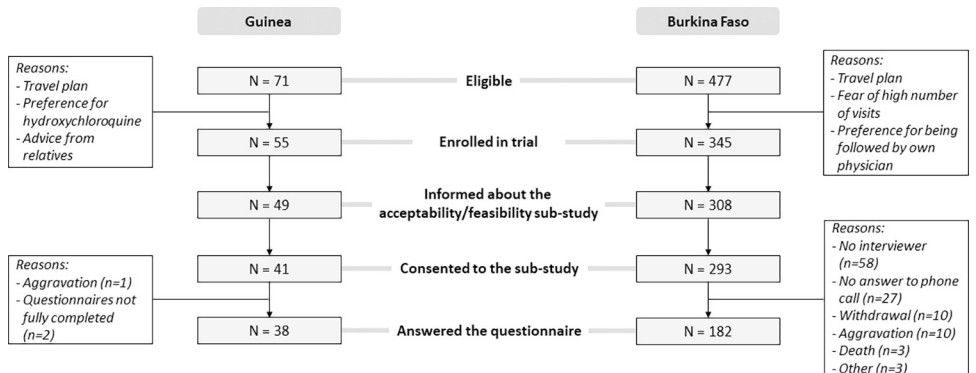

**Fig 1. Population selection for the acceptability / feasibility sub-study of the Coverage-Africa trial.** 2021–2022.

Among them, 220 completed the questionnaire (182 in Burkina Faso and 38 in Guinea) (see description in Table 1); and 24 patients were also interviewed (16 in Burkina Faso and 8 in Guinea) (see characteristics in Table 2).

All participants responding to the questionnaire in Burkina Faso were based at home from trial enrolment; 2 of them were subsequently hospitalized before D8 because of symptoms worsening; at D8, 114 reported having had at least one follow-up visit at home since enrolment. In Guinea, all participants were hospitalized in a 'COVID-19 treatment center' at trial enrolment; at D8, 11 were back home among which 4 reported having received at least one follow-up visit at home (Table 1). As per the participants who were selected for the interviews, all were based at home for the entire duration of their trial participation in Burkina Faso. In Guinea, all were hospitalized for at least one week and 7/8 had returned home before D14. In addition, 11 participants in Burkina Faso and 5 in Guinea had at least one home-based visit before D14-D21 (Table 2).

## Overall acceptability and feasibility of follow-up models

More than 90% of participants reported being "completely satisfied" or "satisfied" overall with the different follow-up models implemented during the trial, either at home or in the hospital, and through complementary phone calls (Fig 2). Participants' satisfaction was however nuanced in the qualitative interviews as well as with specific quantitative statements.

## Acceptability and feasibility of home-based follow-up

**Health considerations.** Participants from both countries generally expressed the view that home was a suitable environment for being followed-up, if one has no or mild symptoms, i.e. that "*the person's physical condition [. . .] does not require 24-hour supervision*" (Burkina Faso, woman, 65 y.o.), and is not worried about a worsening health condition. This positive opinion was supported by–and even conditional on–the reassurance that they were also followed-up by phone during the days without home visits, and that they could call the medical team at any time in case of health concerns. No participant reported having difficulties with taking the treatment at home, either by inhalation or orally. Only a few respondents (12%) were worried about not being able to see a healthcare professional every day during their home follow-up (Fig 2); for example, in his interview, one participant from Burkina Faso was convinced that his level of illness would have required hospitalization and close monitoring.

**Table 1. Description of Accept questionnaire respondents.** Coverage-Africa. 2021–2022.

| | Burkina Faso (n = 182) | | Guinea (n = 38) | | Total (n = 220) | |
|---|---|---|---|---|---|---|
| | N | (%) | n | (%) | N | (%) |
| **Age** (in years) | | | | | | |
| 18–39 | 17 | (9) | 9 | (27) | 26 | (12) |
| 40–59 | 127 | (70) | 14 | (42) | 141 | (66) |
| 60+ | 38 | (21) | 10 | (30) | 48 | (22) |
| *Missing* | *0* | | *5* | | *5* | |
| **Sex** | | | | | | |
| Women | 97 | (53) | 23 | (61) | 120 | (55) |
| Men | 85 | (47) | 15 | (39) | 100 | (45) |
| **Level of schooling** | | | | | | |
| None / Primary | 13 | (7) | 11 | (29) | 24 | (11) |
| Secondary / Professional | 85 | (47) | 8 | (21) | 93 | (42) |
| University | 84 | (46) | 19 | (50) | 103 | (47) |
| *Missing* | *1* | | *0* | | *1* | |
| **Family structure** | | | | | | |
| Lives alone | 2 | (1) | 0 | (0) | 2 | (1) |
| Lives with only one other adult | 11 | (6) | 2 | (5) | 13 | (6) |
| Lives with 2+ other adults without children | 29 | (16) | 9 | (24) | 38 | (17) |
| Lives with at least 1 child | 140 | (77) | 27 | (71) | 167 | (76) |
| **Type of habitation** | | | | | | |
| Individual compound, isolated house | 150 | (82) | 19 | (50) | 169 | (77) |
| Shared compound, apartment block | 32 | (18) | 19 | (50) | 51 | (23) |
| **Treatment administration method** | | | | | | |
| Inhalation | 88 | (51) | 13 | (34) | 101 | (46) |
| Tablets (orally) | 94 | (48) | 25 | (66) | 119 | (54) |
| **Location until D8** | | | | | | |
| Home only | 180 | (99) | 0 | (0) | 180 | (82) |
| First hospitalized then discharged at home before D8 | 0 | (0) | 11 | (29) | 11 | (5) |
| First at home then hospitalized | 2 | (1) | 0 | (0) | 2 | (1) |
| Hospitalization only | 0 | (0) | 27 | (71) | 27 | (12) |
| **Home-based face-to-face follow-up at D7** | | | | | | |
| Yes | 114 | (63) | 4 | (11) | 118 | (54) |
| No | 68 | (37) | 34 | (89) | 102 | (46) |

*"In my condition, at least on the first day, they should have kept me, monitored me, reassured me, so that I could really know that [. . .] there were people who could help me at any time, but that was not the case."* (Burkina Faso, man, 44 y.o.)

**Concerns related to the entourage.** The large majority of participants (92%) were afraid of transmitting COVID-19 to their family (Fig 2), and many of those who participated in the qualitative interviews stated they appreciated the advice given by healthcare providers on how to avoid contamination of family members and neighbors.

Moreover, about 25% of participants in Burkina Faso and 45% in Guinea reported being concerned that their neighbors would know they were sick from COVID-19 when followed at home (Fig 2): in the qualitative interviews, participants explained that home visits by medical teams usually trigger curiosity from neighbors–even gossip or indiscretions–, especially when living in a shared compound.

**Table 2. Description of Accept interviewees.** Coverage-Africa. 2021–2022.

| | Burkina Faso (n = 16) | Guinea (n = 8*) | Total (n = 24) |
|---|---|---|---|
| **Age** (in years) | | | |
| 18–39 | 3 | 1 | 4 |
| 40–59 | 11 | 3 | 14 |
| 60+ | 2 | 4 | 6 |
| **Sex** | | | |
| Women | 7 | 5 | 12 |
| Men | 9 | 3 | 12 |
| **Level of schooling** | | | |
| None / Primary | 0 | 3 | 3 |
| Secondary / Professional | 5 | 0 | 5 |
| University | 11 | 5 | 16 |
| **Type of habitation** | | | |
| Individual compound, isolated house | 13 | 4 | 16 |
| Shared compound, apartment block | 3 | 4 | 7 |
| **Treatment administration method** | | | |
| Inhalation | 3 | 10 | 13 |
| Tablets (orally) | 5 | 6 | 11 |
| **Location until D14-D21 interview** | | | |
| Only at hospital | 0 | 1 | 1 |
| Only at home | 16 | 0 | 16 |
| First hospitalized then discharged at home between the first and the second interview | 0 | 7 | 7 |
| **Home-based face-to-face follow-up at D7 or D14** | | | |
| Yes, at least one | 11 | 5 | 16 |
| No | 5 | 3 | 8 |

* In Guinea, one participant was unable to participate in the 2nd interview on D14-21 due to a worsening of his health condition

*"[. . .] where I live, no one knows that I have COVID-19. [If doctors came to my home] it could make my neighborhood aware that I am sick."* (Guinea, woman, 32 y.o.)

*"[. . .] you know, we are in a society where most of the time if you see a car arriving at someone's place, people will ask themselves: "Ha! Why did they come to visit him?" But me, personally I don't mind because I was able to explain to people that it's done as part of my treatment monitoring, that's why [the caregivers] came to visit me."* (Guinea, man, 45 y.o.)

**Comfort and practicality.** Participants appreciated being at home while being sick from Covid-19 (Fig 2); in the interviews, they reported that they felt more at ease being in their own personal environment and it allowed them to do their household chores. Some women explained that staying at home while they had COVID-19 allowed them to take care of their children–and thus have peace of mind.

*"The fact that we are at home, especially as a mother, makes me feel better, more reassured. At least I can see what is going on at home."* (Burkina Faso, woman, 48 y.o.)

Most participants (96%) perceived that home follow-up visits were convenient (Fig 2). As explained in the interviews, they could indeed avoid the burden of going to the hospital (i.e.

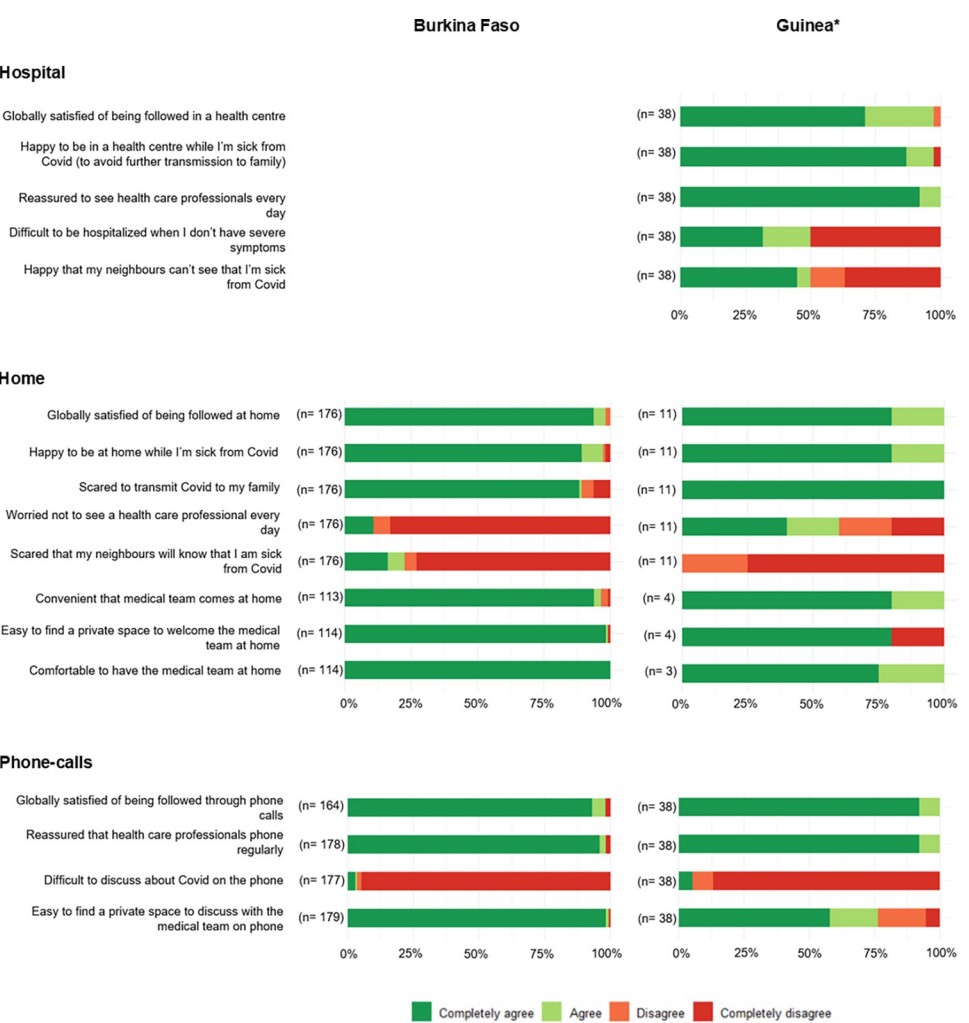

* 11 participants from Guinea were discharged home at D8

**Fig 2. Participants' experiences with the hospital-based, home-based and phone calls follow-up modalities at D8 within the Coverage-Africa trial.** 2021–2022.

travel time; expenses for fuel) and the constraints and inconveniences of being in the hospital (e.g. completing all the administrative procedures; waiting time).

> "I prefer [that the medical follow-up team comes to my home] because it is less effort for me, with the traffic in Conakry, there you come and find me at home, you do your work and leave". (Guinea, woman, 71 y.o.)

In both countries, all participants felt comfortable having the medical team in their home, even if two persons (one in Burkina Faso, one in Guinea) reported that it was not easy to find a private space for the consultation (Fig 2).

**Continuity of care.** For participants from Guinea who were initially hospitalized, sustained follow-up at home after hospital discharge was seen as a continuity of care. In both countries, participants expressed that home visits were an indication that healthcare workers were serious and took care of their patients. Some said home visits made them feel important,

as it is unusual to have a visit from a doctor *"if you are not a boss or a businessman"* (Guinea, man, 64 y.o.). Home visits by the medical team were also seen by some as a novelty, an exceptional innovation that deserves to be encouraged.

*"Following a patient in a hospital, [. . .] declaring him cured and coming to visit him at home, asking about [his] reality, I think that this is an exceptional professionalism, it is an innovation in health that must be encouraged. [. . .] it shows the professionalism of the medical team because we are the focus of their interest".* (Guinea, man, 64 y.o.,)

## Acceptability and feasibility of hospital-based follow-up

**Health considerations.**   In Burkina Faso, participants shared that if they had experienced more severe symptoms, they would have been in favor of being followed-up as inpatients. In Guinea, several participants stated they appreciated having been hospitalized and followed in a treatment center during the infectious phase of their disease. In both countries, participants expressed a sense of reassurance that, in a hospital, there would be increased monitoring and quicker intervention by health professionals if their health deteriorated; in the questionnaires, all the Guinean participants (100%) declared that seeing health care professionals every day was reassuring.

*"[. . .] if it's necessary and [the doctor] feels that my condition requires internment [hospitalization], there's no reason for me to refuse. In my case, that was not the case."* (Burkina Faso, man, 68 y.o.)

*"I think the best thing for me is to stay in the hospital until the test is negative. Because nothing reassures me about those who are followed at home."* (Guinea, man, 45 y.o)

However, 50% of participants from Guinea agreed that it was difficult to be hospitalized for COVID-19 while they were not having severe symptoms. As explained in the individual interviews, some participants perceived that their health condition did not require hospitalization.

*"I'm here, I don't suffer from anything, nothing hurts me, I'm only here to waste time."* (Guinea, woman, 32 y.o.)

**Concerns related to the entourage.**   In the interviews, hospitalization was often perceived as a good strategy to prevent contamination of other family members; this was also reported by 37/38 of questionnaire respondents in Guinea (Fig 2). Interviews also suggested that, in the Guinean context, patients perceived hospitalization as advantageous because the neighborhood would not know that they had COVID-19; half of the respondents to the questionnaire agreed with this statement (Fig 2).

*"[. . .] no one around me knows that I am ill with COVID-19 and hospitalized at the treatment center, for them I have traveled, I am in Kankan."* (Guinea, woman, 32 y.o.)

Some participants were however unhappy with hospitalization. Not being able to get out from the facility was seen as a barrier to managing family life and providing basic care for their children, leading to feelings of worry and frustration. Mandatory hospitalization as in Guinea was even experienced by some participants as being in a prison from where they wanted to leave as rapidly as possible. In addition, the ban on visitors during the COVID-19 pandemic

led hospitalized patients to feel anxious, deprived from customary visits by at least one family member for moral support and care.

*"I had no choice [to be followed in a hospital]. [. . .] my father [. . .] told me to be brave until the day I was released. But this bothers me, as I'm the head of the family. I left my child [who is] ill, I had to borrow money to give to my mother so that she could send my child to a hospital. Every morning I have to go out to get food for my children, otherwise they can't have food. I'm here, I don't suffer from anything, nothing hurts me, I'm here only to waste time [. . .] for nothing. [. . .] I am really upset. [. . .] My mother called me [to tell me] that my child is lying sick, that he even had a seizure. I explained this to the doctor who [. . .] told me to wait. I waited until the evening, [and] I escaped."* (Guinea, woman, 32 y.o.)

*"I'm losing my time [. . .] it's not a place for leisure so I want to run away and get out."* (Guinea, man, 64 y.o.)

**Hospital environment considerations.**   While some patients were totally satisfied with hospitalization–to the point of making one of them "forget the disease" (Guinea, man, 64 y.o.), other patients shared difficult experiences of being in dirty and/or overcrowded rooms.

*"I told Dr [X] that I can't live in this, I can't leave my home, come, and you put me in this. If that's it, I would rather stay at home and confine myself there. At home, I can't say I have luxury, but still, I know it's clean. If I'm sick, I can't come and live where it's dirty.* (Guinea, woman, 55 y.o.)

In Burkina Faso, some participants mentioned that hospital settings were perceived as poorly organized, providing inadequate quality of care at the beginning of the pandemic, and were thus psychologically disruptive to COVID-19 patients (and their families). One participant even feared that hospitalization would equate to death.

*"[. . .] the information and images that we saw at the beginning [of the pandemic] of COVID cases that were interned [i.e. hospitalized], are not very positive. When we see that, we don't expect to have all the care. So I wouldn't want to be [hospitalized] for the study. Because we get the impression that the care is not correct."* (Burkina Faso, woman, 40 y.o.)

*"All of those I knew who were hospitalized, none of them survived. They all passed away. In my opinion they kill them on purpose. It looks like the doctors are giving these patients injections to kill them. It is for this reason that I do not like to go to the hospital."* (Burkina Faso, woman, 43 y.o.)

## Acceptability and feasibility of phone follow-up

**Health considerations.**   In both the interviews and in the questionnaire (Fig 2), phone calls were mainly perceived as reassuring, as participants would be quickly assisted in case of health deterioration, especially for patients followed at home. Participants underlined the importance of combining telephone calls with face-to-face assessments so that the medical team could assess their actual health status. Only a few people (<4%) found it difficult to talk about their illness over the phone (Fig 2).

*[The phone calls] are always a relief, it makes me feel better, it shows that I'm being well followed."* (Burkina Faso, man, 46 y.o.)

*"[. . .] through phone calls they can't see my blood pressure and so, if they come to my house, they can see my condition concretely."* (Guinea, man, 45 y.o.)

**Practicality.** The majority of participants (>99% in Burkina Faso and 76% in Guinea) reported that it was easy to find a private space to discuss on the phone with the trial medical team. Yet, in Guinea, some interview respondents found it difficult especially when they were hospitalized in a shared room, or in the presence of family members or visitors at the time of the calls. Additionally, some spoke of missed calls, unstable telephone network, unavailability of healthcare team members to answer their calls, and calls being somewhat disruptive.

*"Well, that's fine. Except that when you're a bit busy and they call, you find that it's a bit of a trick [disturbing]. . . Otherwise, when you're sitting down, you have nothing to do, and then they call you, [. . .] it is not a problem."* (Burkina Faso, woman, 39 y.o.)

Participants perceived that phone follow-up also presented several advantages for the medical team; it saved travel time especially when workload was too heavy and in the urban context of Conakry where traffic jams are a daily concern.

*[. . .] we are in a city where traffic jams are legendary and the doctor also has his time, but, if we can use this other channel to treat, maybe we will not only save time but, they will also consult many more patients than by [home visit]. I think it's an excellent thing".* (Guinea, man, 64 y.o.)

**Continuity of care.** In the interviews, many participants considered phone calls as a way to ensure continuity of care with face-to-face assessments. Phone follow-up meant that healthcare providers cared about their health and well-being. For some participants, telephone follow-up was perceived as a modern model of care, a technological evolution that can be considered.

*« [. . .] it is the health professionals [. . .] who contributed to having good results by taking the parameters, by giving me the medication. So if the same people after my recovery continue to call me, [. . .] I consider that I am not abandoned to myself. I am followed by professionals and I know that my health is their concern."* (Guinea, man, 64 y.o.)

## Discussion

The COVID-19 pandemic has led to rapid, alternative, innovative, and sometimes drastic decisions in terms of patient management and follow-up, in response to national priorities and strategies for epidemic control. It has also led to considering alternative models for implementing clinical trials, including home-based and hybrid trials, that can include home visits and remote follow-up through the use of mobile technologies [32]. Our findings inform on the acceptability and feasibility of home-based and hospital-based follow-up among pauci-symptomatic COVID-19 patients, in the context of a COVID-19 clinical trial in sub-Saharan Africa. In our study, participants from Burkina Faso and Guinea were satisfied overall with the follow-up implemented in the Coverage-Africa trial–whether they were seen by the trial medical team at home or during hospitalization. They nevertheless raised some issues, mainly relating to the place where they experienced their illness; however, they did not talk much about the follow-up as such, the two being intertwined in their reality. Our analysis showed that factors influencing their views and experiences of the appropriateness of follow-up were sometimes

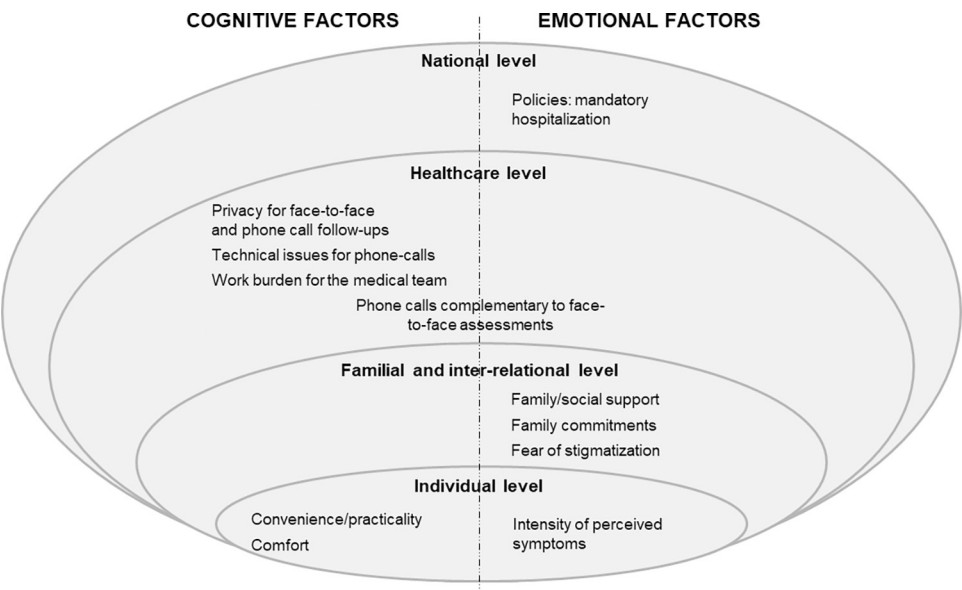

**Fig 3. Factors associated with acceptability follow-ups models of care mapped according to the adaptation of the socio-ecological model.** Coverage-Africa trial. 2021–2022.

cognitive/rational in nature, but more importantly, emotion-related, with recurrent reference to fears/worries or reassurance (Fig 3). Also, participants' concerns pertained to individual, familial/inter-relational, healthcare and national level factors. Finally, participants' views were modulated by the sociocultural context in which they live as well as by the COVID-19 national recommendations.

## Cognitive responses

Convenience/practicality was a key individual-level factor associated with participants' perception of acceptability and feasibility of follow-up. Patients followed-up at home appreciated avoiding the burden of going to the hospital (i.e. travel time; expenses of fuel), as reported in a study conducted in the UK [18]. Patients also shared their satisfaction with being in their own environment, which they perceived as more comfortable compared to the hospital, as also reported for other health conditions in the UK, South Korea and Australia [18–20].

As per factors having an incidence on the organization of healthcare, participants followed at home did not report specific challenges with the feasibility of finding a private room/space for face-to-face assessments and phone follow-ups. However, hospitalized patients in a shared room reported problems finding privacy for phone follow-ups. Technical difficulties such as missed calls or unstable network were among the most common challenges of phone follow-up, as already well-evidenced in the field of digital health [33]. But participants also acknowledged that phone follow-ups allowed the medical team to avoid travel and traffic congestion, thereby reducing their burden–i.e. the perceived effort–related to patient follow-up. In addition, in Guinea, home-based follow-up after hospitalization was perceived as continuity of care, which is consistent with reports from trial participants in the UK [18].

## Emotional responses

Participants' perception of the severity of their COVID-19 symptoms emerged as the most important criterion for judging the appropriateness of the follow-up models in the context of

their participation in the Coverage-Africa trial. Indeed, patients who perceived their symptoms as mild found it appropriate to be followed at home, as long as they were reassured through the phone follow-up and the possibility to contact the medical team by telephone if necessary. In contrast, the hospital setting was considered more reassuring–and therefore more acceptable–for patients with moderate to severe symptoms, in that they could benefit from more rapid assessment and care if their health deteriorated. This need for reassurance stems from the fear of physical and psychological suffering and the fear of dying, which are related to the need for safety–which is recognized as a basic human need according to Maslow's hierarchy [34].

Several participants alluded to both family support and family commitments as key drivers of the acceptability of different follow-up models. Among hospitalized participants, the inability to have a family member at their bedside was a source of distress, as also showed in a study conducted in Austria among hospitalized patients with severe symptoms [17]. Social support is indeed important in times of hardship, especially in the case of emerging epidemics [35, 36], as well as in the West African collectivist context of strong solidarity [37]. Hospitalized mothers also expressed worries and stress about being unable to manage their household and leaving their young children without care. Correspondingly, mothers who were followed-up at home appreciated being able to look after the well-being of their family. In settings where there are no or limited social safety nets, parents–and especially mothers–must rely on their own means to take on their daily lives and, in some cases, to ensure their survival and that of their children [25].

As per inter-relational factors, the acceptability of home-based follow-up was influenced, for some participants by the fear of stigmatization–particularly among those living in conditions where they could easily be observed by the neighborhood (e.g. shared compound). Although stigma and ostracism appear to be an adaptive response to allow uninfected individuals to avoid contracting the COVID-19 disease [38, 39], this social exclusion can cause shame and affect the psychological well-being of those who experience it [40, 41], especially in the West African collectivist context where social ties are central [42–45]. In Guinea, our study carried out with community members showed that some people prefer to avoid disclosing that they have been infected in order to prevent any possible stigma [37]. Hospitalization may be a better option for those patients who wish to remain discreet about their SARS-CoV-2 infection, compared to home follow-up. Nevertheless, others participants, especially in Burkina Faso, appreciated that the trial medical team came to their homes and took into account their physical living environment by giving advice on how to prevent contamination of relatives and neighbors. Overall, our findings suggest that the individual's immediate social environment (e.g. shared vs individual compound) should be part of the considerations for successful health interventions.

Finally, a national level factor emerged from the narratives in Guinea, where hospitalization of all COVID+ individuals in an "epidemiological treatment center" was mandatory. In this context, some participants experienced this obligation as imprisonment; cases of "escape" from these treatment centers were reported. We hypothesize that such reactions stem, among other things, from the social context of mistrust of the Guinean government and health authorities, as reported by community members in Conakry [37]. These findings suggest that policies involving mandatory hospitalization of COVID-19 patients might have led, and could lead to in the future if replicated, some people to avoid being tested for COVID-19, thus preventing them from receiving appropriate care and risking contaminating their family/entourage, which could be counterproductive in terms of epidemic control.

### Strengths and limitations of the study

To our knowledge, this study is the first to explore the acceptability and feasibility, from the perspective of West African participants, of different follow-up models for patients with mild

to moderate infectious disease in the context of their participation in a clinical trial. This study used a mixed-methods design that included, among others, implementation research and social science concepts and research tools. The qualitative sample was characterized by a selection of participants with diversified profiles in terms of age, sex and level of schooling, which allowed us to elicit a variety of points of view. The findings showed a convergence between qualitative and quantitative data, thus contributing to the validity of our results. The fact that the study was conducted in two West African countries allowed us to explore two different local contexts and national policies and sociocultural specificities. Our multidisciplinary team of social science researchers, public health experts, medical doctors (including infectious diseases specialists) and epidemiologists from Burkina Faso, Guinea and France, contributed to enhancing the richness of data collected and to validating the scientific and sociocultural interpretations of the data. Moreover, we used an in-depth analysis methodology, which was based on a comprehensive conceptual framework, allowing for a thorough investigation of the issues at stake.

Yet, we acknowledge some limitations regarding this study. We included only individuals who already consented to participate in the trial, i.e. de facto had accepted the models of follow-up proposed in the trial (aligned to that of the country). In Guinea, national recommendations at the time of the trial—i.e. all individuals who tested positive for COVID-19 had to be hospitalized—may have caused some people to avoid getting tested [37]: this could potentially affect the profile of the patients enrolled in the trial if the latter had different characteristics compared to the patients included. Also, one of the documented reasons for refusal to participate in the Coverage-Africa trial was the fear of a high number of follow-up visits and calls; therefore, the fact that participants in the 'Accept study' had already agreed to take part in the trial and therefore to these follow-up conditions, suggests a possible selection bias leading to higher acceptability. Many participants had a "privileged" profile, i.e. living in a home with a private yard or having a relatively high level of education, which may explain why few of them reported fearing stigmatization; this potential socioeconomic bias related to participant selection may also have prevented the identification of structural barriers regarding home and phone-based follow-up, such as access to individual rooms. Several trial participants who consented to participate in the 'Accept' sub-study did not ultimately complete the questionnaire. This was mainly due to the fact that there was no available research assistant in Burkina Faso for some time; also some participants did not answer the phone calls from interviewers. However, as these non-respondents did not differ from the other participants in terms of sex and age, we believe there is a low risk of selection bias. In addition, although the number of participants included in Burkina Faso is relatively high, it is quite small in Guinea due to the epidemic curve (i.e. low number of confirmed cases at the time of the study), which limits the estimation of observations regarding the quantitative component in this country. Moreover, the questionnaire was drafted in a very short period of time, in the context of an epidemic emergency where the research protocol and tools had to be quickly submitted for ethical approval; some questions may have been too simplistic. This may explain the lack of diversity and contrast in some responses, and thus consequently that we were unable to perform statistical tests to determine whether certain socio-demographic factors were associated with participants' perceptions. As stated above, we have not assessed the experiences of being followed-up in outpatient settings in Burkina Faso; this modality would benefit from evaluation to inform future clinical trials. Also, we have used Sekhon's Theoretical Framework of Acceptability only at the analysis phase and not specifically for the development of the questionnaire and the interview guide: this may explain why we did not find results for all constructs of acceptability (as shown in the S1 Table). Both quantitative and qualitative data were collected by phone; this may have impaired the rapport between the interviewers and the interviewees thus creating an

environment less conducive to the free expression of experiences and feelings [46]. Participants may have minimized their experiences/points of view about their participation in the trial because of social desirability towards the interviewer; however, the interviewers were independent of the medical team so we are confident that this prevarication bias was limited. In addition, patients mostly reported views about the place where they were followed-up rather than their views about the follow-up by healthcare professionals per se; this may have led to some confusion, especially in Guinea where all patients were hospitalized at the onset of their infection, as interactions with healthcare professionals who were not directly involved in the trial might have influenced their perspectives and experiences of the trial follow-up. Finally, the study was conducted in two urban settings: results might not be fully applicable to West-African rural settings.

## Conclusions

The COVID-19 pandemic "catalyzed" the implementation of out-of-hospital follow-up models for clinical trials, such as home and telephone follow-ups. Our study shows that home follow-up in the context of a clinical trial was acceptable for mildly ill West-African patients, mainly provided that (i) they did not perceive their illness as too severe, (ii) it was combined with telemedicine, and (iii) risk of stigma could be avoided. The implementation of strategies to decrease the risk and/or fear of stigma and help communities to better deal with emerging infectious diseases is key to ensure good follow-up conditions for patients and trial participants. Furthermore, home follow-up was generally preferred, especially by mothers who could continue to care for their children. Our results also highlight the importance of emotions in the acceptability of trial follow-up models and suggest that emotion-based considerations should be included when assessing the feasibility and planning the implementation of a trial, or developing any public health strategy. Furthermore, not taking into account emotional imperatives may have an incidence on patients' mental health–by generating worries, stress, distress, anxiety, shame, etc.–which is counterproductive in the context of healthcare, and even deleterious. The results of this study highlighted the importance the importance to conduct formative research on patients' perspectives, concerns and needs according to their living contexts, when planning clinical trials, especially in the context of an epidemic emergency. Also, the recommendations of this study will inform health policy makers on the acceptability and feasibility of models for following patients who are not severely ill, as well as researchers for the design of clinical trials to evaluate the effectiveness of treatments.

## Supporting information

**S1 Table. Participants' subjective assessment of the appropriateness of the different follow-up models of the Coverage-Africa trial, mapped according to the Theoretical Framework of Acceptability (Sekhon et al.,2017). 2021–2022.**
(DOCX)

## Acknowledgments

We would first like to thank all study participants as well as the medical teams involved in the follow-up of participants in Burkina Faso and Guinea. We also would like to thank Dr. Matthias Borchert for feedback on protocol and interview guide development. We are grateful to the ANRS | MIE, sponsor of the ANRS COV33 Coverage-Africa trial, as well as to DNDi, the leading institution of the ANTICOV study consortium and more specifically to its coordinator Dr. Nathalie Strub-Wourgaft, for their support. We would also like to thank Dr. Aminata

Bagayoko who led the fieldwork in Conakry from the onset of the project until August 1st, 2021, as well as to Dr. Camille Fritzell who acted as the international project manager until October 2021.

## Author Contributions

**Conceptualization:** Mélanie Plazy, Marie-Hélène Doucet, Joanna Orne-Gliemann.

**Data curation:** Mélanie Plazy, Marie-Hélène Doucet, Christine Timbo Songbono, Anselme Sanon, Bamba Issiaka.

**Formal analysis:** Mélanie Plazy, Marie-Hélène Doucet.

**Investigation:** Christine Timbo Songbono, Anselme Sanon, Bamba Issiaka.

**Methodology:** Mélanie Plazy, Marie-Hélène Doucet, Joanna Orne-Gliemann.

**Project administration:** Caroline Martin, Anthony L'hostellier.

**Software:** Mélanie Plazy, Marie-Hélène Doucet.

**Validation:** Joanna Orne-Gliemann.

**Writing – original draft:** Mélanie Plazy, Marie-Hélène Doucet.

**Writing – review & editing:** Christine Timbo Songbono, Anselme Sanon, Bamba Issiaka, Caroline. Martin, Inès Da, Anthony L'hostellier, Olivier Marcy, Denis Malvy, Armel Poda, Alexandre Delamou, Abdramane Berthé, Joanna Orne-Gliemann.

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
