## [Decision Letter · Decision Letter 0]

6 Mar 2023

PGPH-D-23-00019

Acceptability and feasibility of home-based and hospital-based follow-up in a COVID-19 clinical trial in Burkina Faso and Guinea: a mixed-method study among patients within the ANTICOV ANRS COV33 Coverage-Africa trial

Dear Dr. Plazy,

Thank you for submitting your manuscript to PLOS Global Public Health. After careful consideration, we feel that it has merit but does not fully meet PLOS Global Public Health’s publication criteria as it currently stands. Therefore, we invite you to submit a revised version of the manuscript that addresses the points raised during the review process.

The manuscript has been evaluated by two reviewers, and their comments are available below.

The reviewers have raised concerns regarding the reporting and methodology of this study. 

Could you please revise the manuscript to carefully address the concerns raised?

We look forward to receiving your revised manuscript.

Kind regards,

Johannes Stortz, PhD

Staff Editor

Journal Requirements:

1. Please send a completed 'Competing Interests' statement, including any COIs declared by your co-authors. If you have no competing interests to declare, please state "The authors have declared that no competing interests exist". Otherwise please declare all competing interests beginning with the statement "I have read the journal's policy and the authors of this manuscript have the following competing interests:"

2. In the online submission form, you indicated that "Study data will not be publicly available. Data can be made available by the sponsor (ANRS) to any researcher interested. Deidentified participant data can be

made available and shared under a data transfer agreement. Requests for access to the Coverage Africa study data should be sent to the corresponding author". All PLOS journals now require all data underlying the findings described in their manuscript to be freely available to other researchers, either 1. In a public repository, 2. Within the manuscript itself, or 3. Uploaded as supplementary information.

Additional Editor Comments (if provided):

Reviewers' comments:

Reviewer's Responses to Questions

**Comments to the Author**

1. Does this manuscript meet PLOS Global Public Health’s publication criteria? Is the manuscript technically sound, and do the data support the conclusions? The manuscript must describe methodologically and ethically rigorous research with conclusions that are appropriately drawn based on the data presented.

Reviewer #1: Partly

Reviewer #2: Yes

2. Has the statistical analysis been performed appropriately and rigorously?

Reviewer #1: Yes

Reviewer #2: Yes

3. Have the authors made all data underlying the findings in their manuscript fully available (please refer to the Data Availability Statement at the start of the manuscript PDF file)?

Reviewer #1: No

Reviewer #2: Yes

4. Is the manuscript presented in an intelligible fashion and written in standard English?

Reviewer #1: No

Reviewer #2: Yes

5. Review Comments to the Author

Reviewer #1: Dear authors,

Firstly, I congratulate the authors for the dedication and focus to fortify the science in public health thematic area. The manuscript is important to the public health and provides high importance to better understanding the acceptability and feasibility of different modalities of follow-up among patients with mild to moderate symptoms enrolled in COVID-19 clinical trials.

I strongly suggest to the authors to follow the PLOS Global Public Health’s submission guidelines. There is no line numbering which difficulted the review of the manuscript.

I suggest reading the manuscript carefully, since there are many little mistakes that could be solved easily by the authors.

The manuscript requires professional editing in the English language to make it easy to read and understand.

I suggest to the authors The Good Reporting of a Mixed Methods Study (GRAMMS) criteria to assess reporting quality of Mixed Method studies. It is available from O'Cathain A, Murphy E, Nicholl J. The quality of mixed methods studies in health services research. J Health Serv Res Policy. 2008;13(2):92-98. And https://www.equator-network.org/reporting-guidelines/the-quality-of-mixed-methods-studies-in-health-services-research/

Follow below the review comments point-by-point to the authors for each section of the manuscript.

Title

I suggest to the author rewrite the title according to PLOS Global Public Health´s submission guidelines.

Abstract

The abstract was well reported.

Introduction

There are two aims in the last paragraph of the introduction. I suggest to the authors to rewrite this paragraph synthetize the main aim of the study. Because it is not clear what was the aim of the study. The authors in the manuscript body reported other aims. Please, clarify it.

Materials and Methods

Replace the word “Methods” to “Materials and Methods” according to PLOS Global Public Health’s submission guidelines.

Why did the authors decide to a mixed methods design?

This is a mixed methods study design. I suggest the authors to organize the headings properly.

Please, organize the level 2 heading in the methods section. Include “Qualitative phase” and “Quantitative phase”. Create the “Data integration” heading.

Where did the integration of the data happened? The authors stated it happened in the interpretation of the results. It is not clear in the methods section how it happened.

It is very important to the reader to understand when the integration of the data and for what it occurred.

Please, organize the level 2 heading in the methods section. Include “Qualitative phase” and “Quantitative phase”. Create the “Data integration” heading.

In the data analysis heading, please insert a reference about “classical thematic analysis”.

I suggest to the authors to clarify the meaning of cultural brokers.

Results

The results were well reported but I suggest to the authors a better integration of the data. The authors stated in the methods section, data analysis heading “Both quantitative and qualitative results were then merged for interpretation”, but it is nor clear where it happened in the results section. The results are reported as single studies.

The results reported are relevant but it is necessary to integrate the data to follow the methodologic procedures of the mixed methods study design.

Discussion

I suggest to the authors a deep literature review about the theme. The authors should discuss the results based in studies in different scenarios.

I suggest to the authors a discussion based in the data from two different paradigms, quantitative and qualitative and how the mixed methods design' findings can subsidize the policy makers in different scenarios to obtain the comprehensiveness of the acceptability and feasibility of home-based and hospital-bases follow-up in a COVID-19 context.

Are there other limitations? For example, in the individual interviews or in the statistical analysis or in the theoretical framework, etc.

Conclusions

The conclusions section was well reported.

I suggest to the authors to insert in the conclusion section a little bit about the impact of the patients’ experiences and perspectives in planning clinical trials, considering the COVID-19 pandemic and the necessity of thinking about the participants in clinical trials as people full of experiences, fears and yearnings. According the results of the study.

References

Replace the word “bibliography” to “References” according to PLOS Global Public Health’s submission guidelines.

Please, insert the DOI number in the published articles when it is available.

Figures

The figures were well reported and provided a better overview about the study, principally figure number 3.

Reviewer #2: Plazy and colleagues conducted a mixed-methods study to evaluate the acceptability and feasibility of follow-up strategies of patients with non-severe COVID-19 recruited in the ANTICOV ANRS COV33 Coverage-Africa trial. Their main conclusions are that hospital-based follow-up was viewed as a way to prevent contamination of family members, but could be badly experienced when mandatory and conflicting with family responsibilities and commitments. Phone calls were seen as reassuring and as a way to ensure continuity of care.

The article is well-written, methods are sound, and the results are presented and discussed overall in a proper manner. Major limitations are adequately acknowledged

I only have a major concern, which refers to the overlap/confusion of follow-up strategies as per guidelines vs. per trial procedures and how this is reflected in the text. In particular for patients treated in Guinea, all of whom were hospitalized in the initial steps of the infection, it is unclear how the interaction with healthcare professionals not directly involved in the trial might have had influenced their perspective. Could you please elaborate on this?

As minor comments:

1. In my opinion the abstract does not reflect all the major findings. For instance, home-based follow-up is not clearly addressed. Or "Our results also highlight the importance of emotions in the acceptability of trial follow-up models and suggest that emotion-based considerations should be included when assessing the feasibility and planning the

implementation of a trial, or developing any public health strategy."

2. Methods, study setting. Please, include a clear definition +/- reference for "non-severe symptoms form

of the disease"

3. "Trial procedures regarding patient enrolment and follow-up were defined in accordance with the COVID-19 national recommendations prevailing during the study." This sentence deserves further detail, in particular regarding whether this might have affected the profile of patients enroled and this, in turn, has affected the follow-up procedures and therefore how participants in the study have perceived them.

4. In the limitations, authors already tackle the potential selection bias by level of education. However, could you please elaborate on the fact that only two patients leaved alone (beside the community-oriented society aspect) and how might have impacted the findings and the potential external validity? Regarding the educational/socioeconomic position, could the bias prevent identifying structural barriers for at home treatment and follow-up such as individual rooms, open air spaces, availability of mobile phones, etc? Could you please elaborate on the relationship between socioeconomic position and fear of stigmatization.

5. Any particular conclusion from the interviews of those patients that received hospitalization only (71% of patients from Guinea)? Is shocking that one of the highlighted statements is: "However, 50% of participants from Guinea shared that it was difficult to be hospitalized for COVID-19 while they were not having severe symptoms. This was also shared in the individual interviews, as some participants perceived that their health condition did not require hospitalization." All participants had non-severe COVID-19 and were all hospitalized.

6. PLOS authors have the option to publish the peer review history of their article (what does this mean?). If published, this will include your full peer review and any attached files.

**Do you want your identity to be public for this peer review?** For information about this choice, including consent withdrawal, please see our Privacy Policy.

Reviewer #1: No

Reviewer #2: **Yes: **Juan M Pericàs

---

## [Decision Letter · Decision Letter 1]

20 Jun 2023

Acceptability and feasibility of home and hospital follow-up in Burkina Faso and Guinea: a mixed-method study among patients of the COVID-19 Coverage-Africa clinical trial

PGPH-D-23-00019R1

Dear Dr Melanie Plazy 

We are pleased to inform you that your manuscript 'Acceptability and feasibility of home and hospital follow-up in Burkina Faso and Guinea: a mixed-method study among patients of the COVID-19 Coverage-Africa clinical trial' has been provisionally accepted for publication in PLOS Global Public Health.

Best regards,

Ferdinand Mukumbang, PhD

Academic Editor

Reviewer Comments (if any, and for reference):

Reviewer's Responses to Questions

**Comments to the Author**

1. If the authors have adequately addressed your comments raised in a previous round of review and you feel that this manuscript is now acceptable for publication, you may indicate that here to bypass the “Comments to the Author” section, enter your conflict of interest statement in the “Confidential to Editor” section, and submit your "Accept" recommendation.

Reviewer #1: All comments have been addressed

Reviewer #2: All comments have been addressed

2. Does this manuscript meet PLOS Global Public Health’s publication criteria? Is the manuscript technically sound, and do the data support the conclusions? The manuscript must describe methodologically and ethically rigorous research with conclusions that are appropriately drawn based on the data presented.

Reviewer #1: Yes

Reviewer #2: Yes

3. Has the statistical analysis been performed appropriately and rigorously?

Reviewer #1: N/A

Reviewer #2: Yes

4. Have the authors made all data underlying the findings in their manuscript fully available (please refer to the Data Availability Statement at the start of the manuscript PDF file)?

Reviewer #1: Yes

Reviewer #2: Yes

5. Is the manuscript presented in an intelligible fashion and written in standard English?

Reviewer #1: Yes

Reviewer #2: Yes

6. Review Comments to the Author

Reviewer #1: Dear authors,

Thank you for revising the manuscript considering the suggestions.

The manuscript is now technically sound.

I do not have additional comments on the revision in this manuscript version.

Reviewer #2: The authors have provided clear and comprehensive responses to my questions and have applied changes to the manuscript that satisfactorily fulfill the remarks I made in my first review.

7. PLOS authors have the option to publish the peer review history of their article (what does this mean?). If published, this will include your full peer review and any attached files.

**Do you want your identity to be public for this peer review?** For information about this choice, including consent withdrawal, please see our Privacy Policy.

Reviewer #1: No

Reviewer #2: **Yes: **Juan M Pericàs
